# Transcriptomic Analysis Provides Novel Insights into Heat Stress Responses in Sheep

**DOI:** 10.3390/ani9060387

**Published:** 2019-06-24

**Authors:** Zengkui Lu, Mingxing Chu, Qing Li, Meilin Jin, Xiaojuan Fei, Lin Ma, Liping Zhang, Caihong Wei

**Affiliations:** 1Key Laboratory of Animal Genetics and Breeding and Reproduction of Ministry of Agriculture, Institute of Animal Science, Chinese Academy of Agricultural Sciences, Beijing 100193, China; luzk0911@163.com (Z.L.); mxchu@263.net (M.C.); liqing_0507@163.com (Q.L.); jmlingg@163.com (M.J.); 18409481571@163.com (X.F.); 18633082661@163.com (L.M.); 2College of Animal Science and Technology, Gansu Agricultural University, Lanzhou 730070, China; zhangliping@gsau.edu.cn

**Keywords:** sheep, heat stress, RNA-Seq, transcriptome

## Abstract

**Simple Summary:**

The general increase in global temperatures has meant that heat stress has become an increasingly significant problem for sheep. This has both direct and indirect impact on their physiological functions, productivity, and health of sheep. Sheep generally live in high-temperature environments; however, the genes and pathways that play regulatory roles in the heat stress responses of sheep remain unclear. In this study, we applied RNA-Seq technology to analyze liver tissues of sheep from heat-stressed and control groups, and screened genes and pathways related to sheep heat stress. This work provides a theoretical foundation for the breeding and production of heat-resistant sheep.

**Abstract:**

With the intensified and large-scale development of sheep husbandry and global warming, sheep heat stress has become an increasingly important issue. However, little is known about the molecular mechanisms related to sheep responses to heat stress. In this study, transcriptomic analysis of liver tissues of sheep in the presence and absence of heat stress was conducted, with the goal of identifying genes and pathways related to regulation when under such stress. After a comparison with the sheep reference genome, 440,226,436 clean reads were obtained from eight libraries. A *p*-value ≤ 0.05 and fold change ≥ 2 were taken as thresholds for categorizing differentially expressed genes, of which 1137 were identified. The accuracy and reliability of the RNA-Seq results were confirmed by qRT-PCR. The identified differentially expressed genes were significantly associated with 419 GO terms and 51 KEGG pathways, which suggested their participation in biological processes such as response to stress, immunoreaction, and fat metabolism. This study’s results provide a comprehensive overview of sheep heat stress-induced transcriptional expression patterns, laying a foundation for further analysis of the molecular mechanisms of sheep heat stress.

## 1. Introduction

Since its foundation in 1988, the Intergovernmental Panel on Climate Change has released five comprehensive climate reports and highlighted the indisputable fact that the world is warming. In the wild, air temperature has a major bearing on the ability of animals to thrive. Long-term high temperatures and excessively warm weather generate serious heat stress for animals. Heat stress directly or indirectly affects animal physiology, production, and health, among others [1,2,3], which is manifested by physiological features such as gasping for breath, accelerated respiration, elevated heart rate, rising rectal temperature, increasing in-vivo oxidative metabolism, and water and electrolyte disturbance [4,5,6,7]. Regarding reproduction, it can also cause reduced food intake, weight loss, and declining fertility [8,9,10], as well as declining immunity and disease resistance [11,12].

Transcriptomics refers to the study of all of the transcripts in cells, tissues, or living organisms at a specific developmental stage or under specific physiological conditions [13]. In contrast to the relatively stable genome, the transcriptome varies among developmental stages, physiological conditions, and upon exposure to external stimuli. The transcriptome, which has become a powerful tool for analyzing the relationship between genotype and phenotype, has been extensively applied to the analysis of complicated biological events.

To date, studies on heat stress in animals have mainly focused on dairy cow, broiler, and pig. These studies have mainly concentrated on features such as physiology and biochemistry, production performance, meat quality, and immunological functions. However, these studies were generally restricted to individual genes and pathways and could not characterize the complicated processes of animal responses to heat stress holistically. Moreover, very few studies have reported on heat stress in sheep. There is thus a need for analyses of the genes and pathways that exert regulatory effects when sheep are under heat stress. Against this background, Hu sheep were selected as experimental samples in this study, and subjected to RNA-Seq, bioinformatic, and molecular biology analyses to explore the molecular mechanisms of sheep heat stress, to seek genes and regulatory pathways related to sheep heat stress, and to provide a theoretical basis for sheep production and breeding practices that will help sheep producers deal with heat stress in their sheep.

## 2. Materials and Methods 

### 2.1. Animals and Sample Collection

All the experimental procedures mentioned in the present study were approved by the Science Research Department (in charge of animal welfare issue) of the Institute of Animal Sciences, Chinese Academy of Agricultural Sciences (IAS-CAAS) (Beijing, China). Ethical approval on animal survival was given by the animal ethics committee of IAS-CAAS (No. IASCAAS-AE-03, 12 December 2016). Thirty-six purebred 5-month-old healthy Hu sheep (half females and half males with approximately equivalent weights. At this age, sheep of this breed have normally achieved puberty) were selected in August (heat-stressed group) and December 2017 (no heat stress, control group) and bred at Jiangsu Qianbao Sheep Industry Co., Ltd. (Yancheng, Jiangsu, China). Sheep in the heat-stressed group and the control group were fed the same food and had free access to water. Five temperature and humidity meters were hung 1.5 m above the pen, and temperature (T_d_) and relative humidity (RH) were recorded every 2 h from 08:00 to 20:00. Temperature-humidity index (THI) was used to evaluate the severity of heat stress [14]. When THI was continuously higher than 23.3 (severe heat stress) or continuously lower than 22.2 (no heat stress) for 7 d, four Hu sheep (half females and half males; HG1 and HG2 were heat-stressed males and HM1 and HM2 were heat-stressed females, while CG1 and CG2 were males in the control group and CM1 and CM2 were females in the control group) were randomly selected and slaughtered on an empty stomach the morning (sheep were fasted for 12 h before slaughter, but had free access to water), after which complete liver tissues were separated out, and some left hepatic lobe tissues were preserved in liquid nitrogen for RNA extraction.

### 2.2. Detection of Thyroid-Related Hormones in Sheep Serum

Blood was collected from the jugular vein on an empty stomach on the day of slaughtering the sheep. Five sheep were selected for this at random in heat stress and control groups, for which non-anticoagulant vacutainer tubes were used to collect 5 ml blood. Serum was then separated from it by centrifugation for 15 min at 3000 r/min. In accordance with the manufacturer’s instructions, radioimmunoassay was used to determine the T_3_ {Iodine [^125^I] 3,3’,5-Triiodothyronine(T_3_) Radioimmunoassay Kit, Beifang, Beijing, China} and T_4_ {Iodine [^125^I] Thyroxine Radioimmunoassay Kit, Beifang, Beijing, China} concentrations in sheep serum.

### 2.3. RNA Extraction and Quality Inspection

Total RNA in the liver tissues of the eight sheep was extracted using TRlzol reagent (Invitrogen, Carlsbad, CA, USA), in accordance with the manufacturer’s instructions. The integrity and concentration of RNA were determined by agarose gel electrophoresis and a Nanodrop 2000 (Thermo, Waltham, MA, USA). Finally, the RNA concentration was accurately quantified using Qubit 2.0 (Invitrogen, Carlsbad, CA, USA) and stored at −80 °C until subsequent analysis.

### 2.4. cDNA Library Construction and Sequencing

RNA samples that qualified through quality inspection were used for library construction in accordance with the instructions provided by Illumina (Illumina, San Diego, CA, USA). The specific steps were as follows: (1) Oligo-carrying magnetic beads were used for mRNA enrichment. (2) Fragmentation buffer was used to randomly break enriched mRNA into fragments of about 200 nucleotides in length. (3) The fragmented mRNA was used as a template; one-chain cDNA was synthesized through inverse transcription using a random primer. In the synthesis of second-chain cDNA, dTTP in dNTPs was replaced by dUTP. (4) AMPure XP beads were used to purify double-chain cDNA, and End Repair Mix was used for end filling and the addition of an A-tail and sequencing linkers. (5) USER enzyme was used to digest double-stranded cDNA so that the library only contained single-stranded cDNA. (6) PCR enrichment of the cDNA fulfilling the above criteria was conducted. (7) Qubit 2.0 and Agilent 2010 (Agilent, Palo Alto, CA, USA) were used for quality inspection of the concentration and fragment size of the established library. The Illumina Hiseq 2500 platform was used to conduct double-end sequencing of the eight established libraries.

### 2.5. Raw Data Preprocessing and Alignment

Raw reads were obtained after Illumina sequencing ended. Skewer software was used to dynamically remove the 3’ ends, linker sequences, and low-mass sequences of raw reads, after which clean reads were obtained. FastQC software was used to conduct quality control analysis of preprocessed data and determine indexes such as GC content, Q20, and Q30. STAR software was used to align obtained clean reads onto reference genomes of sheep (Oar_v4.0), where alignment parameters were set as —twopassMode Basic, —outSAMstrandField intronMotif, —alignSJstitchMismatchNmax 5, −1, 5, 5, and other parameters were set as the defaults. Those reads aligned to more than one (>1) genome were removed and the others were kept for subsequent analysis.

### 2.6. Identification and Analysis of Differentially Expressed Genes

StringTie software was used for counting the original sequences of known genes of the eight samples. Expression levels of known genes were expressed using fragments per kilobase of transcript per million fragments mapped (FPKM), where FPKM = Total fragments/Mapped reads (millions) × Exon length (KB). The genes were divided into eight intervals [(0.1–1), (1–5), (5–10), (10–20), (20–30), (30–40), (40–50), and (≥50)] according to their FPKM values, and the number of transcripts in each interval and its percentage relative to the total number of transcripts were calculated. It is generally considered that FPKM ≥ 0.1 indicates that the transcript is expressed. DESeq2 software was used to screen out genes that were differentially expressed between sheep in the heat-stressed group and those in the control group; in the screening, the thresholds of *p*-value ≤ 0.05 and fold change ≥ 2 were applied. Clustering analysis of the screened differentially expressed genes was subsequently carried out.

### 2.7. Functional Enrichment Analysis of Differentially Expressed Genes

The DAVID database was used in Gene Ontology (GO) functional enrichment analysis of the differentially expressed genes. All genes were placed on the background list; the differentially expressed genes were included in the candidate list, and hypergeometric distribution was used to calculate *p* values before multiple testing and Benjamini–Hochberg correction. GO items with a false discovery rate ≤ 0.05 were regarded as significantly enriched. The Kyoto Encyclopedia of Genes and Genomes (KEGG) database was used in the functional annotation and classification of genes associated with differential peaks.

### 2.8. Verification of Differentially Expressed Genes

The RNA-Seq results were verified using qRT-PCR. A cDNA Synthesis Kit (TransStart Green, Beijing, China) was used for RNA reverse transcription. Oligo 7 software was used for cross-exon design of all primers and specific detection in NCBI. The qPCR SuperMix (TransStart Green, Beijing, China) and LightCycler 480 apparatus (Roche, Basel, Switzerland) were used for quantitative real-time PCR (qRT-PCR). Each sample was measured in triplicate to ensure the accuracy of the quantification. The relative expression of target genes was calculated by the 2^-ΔΔCt^ method using β-actin (*ACTB*) as an internal reference gene. Information on the primers used in this study is listed in Appendix A.

### 2.9. Statistical Analysis

All data reported are expressed as mean ± SE. Student’s t-test was carried out using SPSS software for statistical analysis of the data. A *p* value of < 0.05 was considered to be statistically significant.

## 3. Results

### 3.1. Influence of Heat Stress on Serum T_3_ and T_4_ Levels of Sheep

During heat stress, THI was ≥ 26.18, indicating that the sheep were exposed to extremely severe heat stress; in the control group, THI ≤ 11.82, indicating that the sheep experienced no heat stress (Appendix A) [14]. T_3_ and T_4_ act in regulating the body’s metabolic heat production. In this study, the levels of T3 and T_4_ in the serum of heat-stressed and control sheep were determined and they were found to be extremely highly expressed in the control group sheep (Table 1).

### 3.2. Summary of Sheep Pen THI and Sequencing Data

The Illumina Hiseq 2500 platform was used to conduct paired-end sequencing of eight established libraries; the results of data quality control are shown in Table 2. A total of 513,487,656 raw reads were generated in the eight libraries, 440,226,436 clean reads were left after quality control, and the average proportion of clean reads was 99.06%. The GC contents of the eight samples were within 49.82%–51.25%, and were thus in accordance with base composition rules. Q20 ≥ 98.10% and Q30 ≥ 94.40%. Clean reads were aligned to the sheep reference genome using STAR software, and over 93.25% of the reads could be accurately aligned with a high matching rate. Approximately 7.05%–12.37% of these clean reads had multiple aligned positions, 81.99%–86.73% of them had single aligned positions, and 3.66%–6.76% of them were not aligned to the sheep reference genome. Those reads that aligned on the sheep reference genome at a single position were used for further bioinformatic analysis.

### 3.3. Analysis of Overall Gene Expression Levels

Through a comparison of the FPKM values of all genes, the overall gene expression levels of each sample could be evaluated. As shown in Figure 1, the overall gene expression levels of the eight samples were similar. To further evaluate the overall gene expression levels, the genes were divided into eight intervals (0.1–1, 1–5, 5–10, 10–20, 20–30, 30–40, 40–50, and ≥50) according to their FPKM values, and the number of transcripts in each interval and its percentage relative to the total number of transcripts were calculated. There were small percentage changes of various expression quantities in the eight samples, and the number of transcripts gradually decreased as the FPKM value increased (Appendix A). Overall, most transcripts were expressed at low levels in sheep liver in the heat-stressed group and control group (Appendix A).

### 3.4. Screening and Clustering Analysis of Differentially Expressed Genes

DESeq2 software was used to screen out sheep genes differentially expressed between the heat-stressed group and the control group; 1,137 genes fulfilling both criteria of *p*-value ≤ 0.05 and fold change ≥ 2 were identified (Figure 2A and Appendix A). Compared with the levels in the control group, 

798 genes in sheep in the heat-stressed group presented upregulated expression, while 339 genes were downregulated (Figure 2B). To obtain a deeper understanding of the gene expression patterns of sheep in the heat-stressed group and control group, clustering analysis of these differentially expressed genes was conducted. The results indicated that the differentially expressed genes of sheep in the heat-stressed group and control group were clustered into a single class (Figure 2C). In addition, genes with upregulated differential expression and those with downregulated differential expression were clustered into one class. We also analyzed the effects of heat stress on gene expression in sheep of different genders. 4,203 genes were found to be differentially expressed between heat-stressed rams and rams from the control group (Appendix A), while 1,369 were differentially expressed between heat-stressed ewes and control group ewes (Appendix A).

### 3.5. Functional Enrichment Analysis of Differentially Expressed Genes

Sheep genes that were differentially expressed between the heat-stressed group and the control group, were significantly enriched in 419 GO terms, including 357 biological processes, 27 cellular components, and 35 molecular functions (Appendix A). These GO terms are associated with stress response, immunoreaction, and fat metabolism, among others (Figure 3A). KEGG pathway analysis found that these differentially expressed genes were significantly enriched in 51 pathways (Appendix A). Through a further analysis, many of these pathways were shown to be related to fat metabolism, such as regulation of lipolysis in adipocytes, MAPK (mitogen-activated protein kinase), and PI3K-AKt (phosphatidylinositol 3-kinase and protein kinase B) (Figure 3B). They were also particularly associated with signaling pathways regulating stress reactions, including TNF (tumor necrosis factor) and Rap1 (Figure 3B). Besides, genes differentially expressed in heat-stressed rams and control group rams were particularly associated with the process of fat metabolism and stress response (Appendix A). The results for genes differentially expressed between heat-stressed ewes and control group ewes were also particularly associated with the process of fat metabolism and stress response (Appendix A).

### 3.6. Validation of RNA-Seq Data by qRT-PCR

To verify the sequencing results of RNA-Seq, five genes with high expression and five with low expression in the heat-stressed group in this study were selected for confirmatory analysis by qRT-PCR. As shown in Figure 4, although their expression levels differed to a certain degree, their expression patterns were identical, thus verifying that the RNA-Seq results were accurate and reliable.

## 4. Discussion

Heat stress impacts animal production and endangers animal welfare, causing an annual economic loss of over US$1.2 billion in animal husbandry globally [15]. In conventional breeding methods, the resistance of animals to heat can only be measured by determining some key production indexes, but the genetic changes of animals in association with environmental changes cannot be determined. In contrast, high-throughput sequencing can be used to shed light on animal heat resistance and genes associated with sensitivity to heat, and it has been proven to be a feasible approach in studies of animals such as cow, pig, and chicken [16,17,18]. Therefore, starting from transcriptomics, this study was aimed at identifying sheep heat resistance-related genes and pathways. Four sheep under heat stress and four without such stress were selected in this study, from which 1,137 differentially expressed genes were screened out using RNA-Seq. qRT-PCR results revealed expression patterns of 10 genes that were identical to those in the RNA-Seq results, proving that the sequencing results were accurate and reliable. These differentially expressed genes were mainly related to stress response, energy metabolism, and immunity. Further functional enrichment analysis of these differentially expressed genes was carried out, which showed that most genes were particularly associated with signaling pathways such as stress response, fat metabolism, and immunity.

### 4.1. T_3_ and T_4_

T_3_ and T_4_ are hormones secreted from the thyroid gland and its peripheral tissues. T_3_ is a type of thyroid hormone that works on anabolism and catabolism. The reduction in secretion of T_3_ is considered to reflect an animal’s efforts to reduce its body heat production and accumulation as well as to maintain its body heat balance [1,19]. T_4_ also plays an important role in the adaptation of animals to environmental changes. It stimulates intracellular oxygen consumption and heat production, thereby increasing basal metabolic rate, increasing glucose utilization, and altering lipid metabolism [19,20]. In this study, it was found that the concentrations of T_3_ and T_4_ in the blood of heat-stressed sheep were significantly lower than those in the sheep of the control group, which was consistent with the results in studies of other sheep breeds [7]. A decrease in the concentrations of T_3_ and T_4_ indicates that heat stress reduces the metabolic activity of the sheep, so as to inhibit the body from producing more heat. In addition, the decrease in thyroxine concentration may also be related to a decrease in blood glucose concentration, which is closely related to energy metabolism during heat stress.

### 4.2. Regulation of Body Temperature

5-Hydroxytryptamine receptor 4 (HTR4) presented significantly high expression in the heat-stressed sheep in this study, while HTR1B was significantly more highly expressed in sheep in the control group. It has been indicated that 5-HT plays a significant role in the process of regulating animal body temperature [21]. When animals are in a high-temperature environment, the 5-HT neuroendocrine system is activated and combines with 5-HTR to activate downstream cyclic adenosine monophosphate (cAMP) and cyclic guanosine monophosphate (cGMP) signaling pathways, which enhance neural activity sensitive to heat so as to inhibit heat production [22]. Fewer studies have been performed on HTR4 with regard to body temperature regulation, but it has been shown in animals to exert important regulatory effects in gastrointestinal sensitivity, memory, and food intake, among others [23]. Under heat stress, sheep exhibited reduced food intake and rumination activities, which further reduced the amount of heat generated by metabolism so as to maintain a stable body temperature [6,9]. During this period, HTR4 might play a significant role. HTR1B is also involved in body temperature regulation, and was shown to be activated to reduce the body temperature of rats [24]. Here, HTR1B presented high expression in sheep in the control group, suggesting its importance in regulating body temperature.

### 4.3. Regulation of Stress Reactions

Under heat stress, animals need to undertake a series of nonspecific responses to maintain homeostasis. The functional enrichment analysis of differentially expressed genes in sheep in the heat-stressed group and the control group showed the significant enrichment of genes in stress-related pathways, such as Rap1, MAPK, and PI3K-Akt. Rap1 is regulated by a wide range of external stimuli as a molecular switch when stress occurs [25]. MAPK is one of the main signaling pathways in mammals and can also be activated by external stimuli, but the activity of the MAPK signaling pathway is stimulated by Rap1 in multiple cells [26,27]. In a similar way, the PI3K-Akt pathway is activated by external stimuli of many cellular types and can involve a cascade reaction with the MAPK signaling pathway [28,29]. In summary, these stress-related signaling pathways can form a complicated cascade reaction to respond to heat stress in sheep, and reduce the harm brought about by such stress.

### 4.4. Regulation of Energy Metabolism

Under heat stress, sheep reduce their food intake in order to reduce metabolic heat production. This study showed that differentially expressed genes in the heat-stressed group were particularly associated with fat metabolism-related pathways. Besides regulating stress reactions, MAPK and PI3K-Akt signaling pathways exert important effects on the fat metabolic process [30,31]. In addition, the cAMP signaling pathway also plays a significant role in the hepatic fat metabolic process and can activate protein kinase A (PKA) to promote β-oxidation of hepatic fat [32]. Genes associated with these signaling pathways were also particularly common among the differentially expressed ones, such as *NPR1* (natriuretic peptide receptor 1), *ANGPT2* (angiopoietin 2), and *SLC13A5* (sodium-dependent citrate transporter member 5); this study indicated that all of these genes participate in the regulation of fat metabolism [33,34,35]. Sheep needed to mobilize a large amount of energy to respond to the heat stress, which enhanced catabolism and reduced anabolism correspondingly. Although genes associated with some glycometabolism-related signaling pathways were also commonly identified in this study, compared with fat, saccharide energy content was low with easy hydration. Therefore, fat mobilization in sheep under heat stress could help them adapt to this condition.

### 4.5. Regulation of Immunoreactions

Moderate heat stress has the effect of promoting immune system activity in animals so that they can better adapt to the environment. However, severe heat stress has a negative influence on the immune systems of animals and results in deterioration of their disease resistance. IL1R (interleukin type 1 receptor) has two subtypes, IL1R1 and IL1R2, both of which exert important effects on immune and inflammatory reactions [36]. HSPA2 belongs to the HSP70 (heat stress protein 70) family. It has been shown that HSP70 can increase the secretion of IL-1β from cells and facilitate the transport of antigens to T cells [37,38,39]. In this study, it was found that IL1R1, IL1R2, and HSPA2 presented high expression in heat-stressed sheep, suggesting that these genes play significant roles in the immunoregulatory process of sheep.

## 5. Conclusions

Upon RNA-Seq analysis of eight libraries of heat-stressed and control groups, 1,137 differentially expressed genes were screened out. These genes are mainly involved in biological processes such as stimulus-response reaction, immunoreaction, and fat metabolism. These candidate genes and pathways should deepen our understanding of the molecular mechanisms involved in heat stress in sheep, and provide a theoretical basis for improved sheep production and breeding of heat-tolerant varieties.

## Figures and Tables

**Figure 1 animals-09-00387-f001:**
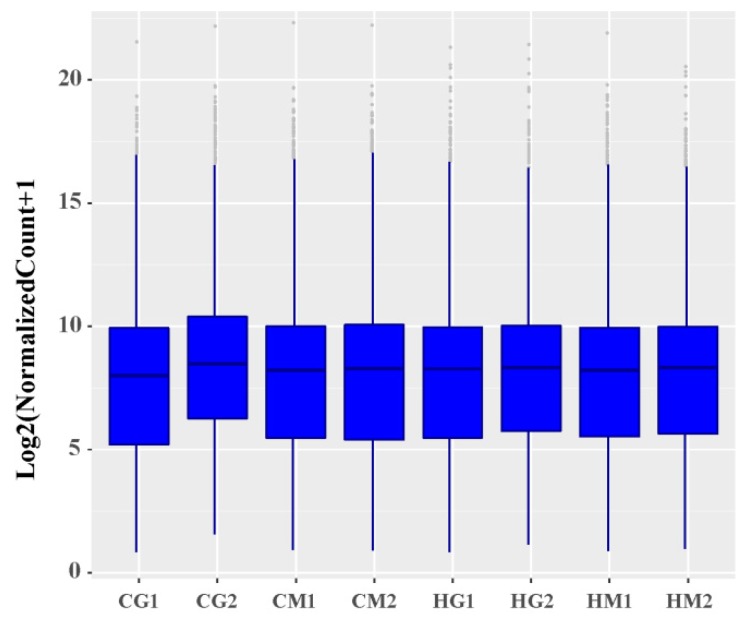
Distribution of gene expression levels. CG1 and CG2 were males in the control group and CM1 and CM2 were females in the control group, while HG1 and HG2 were heat-stressed males and HM1 and HM2 were heat-stressed females.

**Figure 2 animals-09-00387-f002:**
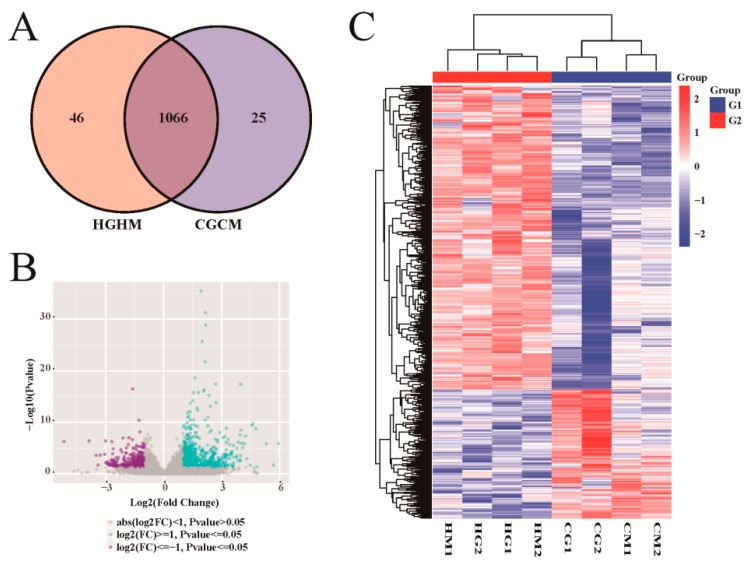
Sheep genes identified to be differentially expressed between heat-stressed and control groups. (**A**) Venn diagram showing the differentially expressed genes. (**B**) Volcanic plot of the differentially expressed genes. (**C**) Heat map of the differentially expressed gene clustering analysis. HGHM were heat-stressed males and females, while CGCM were males and females in the control group.

**Figure 3 animals-09-00387-f003:**
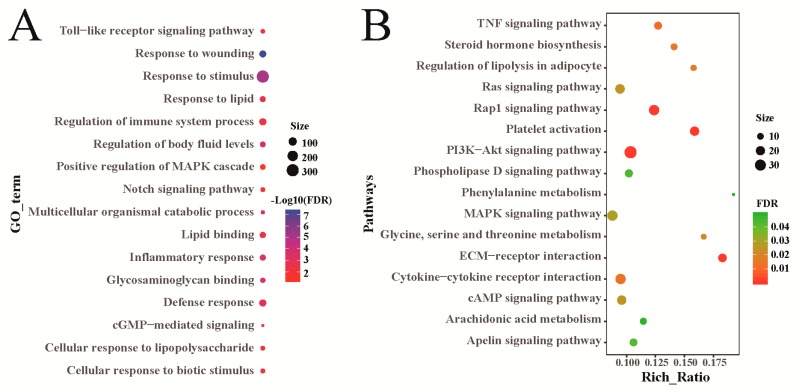
Functional analysis of sheep genes differentially expressed between the heat-stressed and control groups. (**A**) GO annotation of the differentially expressed genes. (**B**) KEGG enrichment analysis of the differentially expressed genes.

**Figure 4 animals-09-00387-f004:**
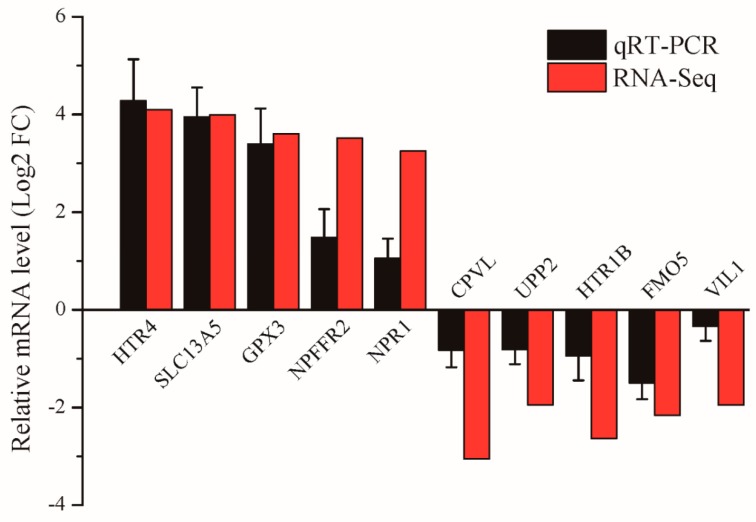
Validation of differentially expressed genes by qRT-PCR.

**Table 1 animals-09-00387-t001:** Influence of heat stress on serum T_3_ and T_4_ concentration of sheep.

Item	Sex	Number	Control Group	Heat Stress Group
T_3_ (ng/mL)	♂	5	3.92 ± 0.041 ^A^	1.24 ± 0.020 ^B^
	♀	5	3.89 ± 0.045 ^A^	1.28 ± 0.016 ^B^
T_4_ (ng/mL)	♂	5	82.92 ± 1.55 ^A^	41.89 ± 1.23 ^B^
	♀	5	76.05 ± 1.17 ^A^	42.73 ± 0.74 ^B^

In the same line, values with different upper-case letters are highly significantly different (*p* < 0.01).

**Table 2 animals-09-00387-t002:** Summary of sequenced RNA-Seq data.

Items	CG1	CG2	CM1	CM2	HG1	HG2	HM1	HM2
Raw reads	69,008,628	39,367,262	69,141,276	68,952,966	78,420,396	51,776,410	76,392,674	60,428,044
Clean reads	68,367,484	38,903,468	68,478,224	68,463,512	77,762,862	51,308,882	75,471,938	59,933,578
Clean ratio (%)	99.07	98.82	99.04	99.29	99.16	99.1	98.79	99.18
GC content (%)	49.82	50.09	51.19	50.92	51.18	51.04	50.81	51.25
Q20 (%)	98.65	98.6	98.5	98.7	98.65	98.5	98.1	98.6
Q30 (%)	95.8	95.75	95.35	95.9	95.8	95.35	94.4	95.55
Total mapped	65,863,115 (96.34%)	36,273,376 (93.24%)	65,257,697 (95.30%)	65,546,575 (95.74%)	73,440,220 (94.44%)	48,208,876 (93.96%)	71,214,523 (94.36%)	56,581,847 (94.41%)
Multiple mapped	6,565,779 (9.60%)	2,743,494 (7.05%)	6,088,334 (8.89%)	6,757,881 (9.87%)	7,483,120 (9.62%)	6,034,386 (11.76%)	9,336,634 (12.37%)	5,753,481 (9.60%)
Unique mapped	59,297,336 (86.73%)	33,529,882 (86.19%)	59,169,363 (86.41%)	58,788,694 (85.87%)	65,957,100 (84.82%)	42,174,490 (82.20%)	61,877,889 (81.99%)	50,828,366 (84.81%)
Unmapped	2,504,369 (3.66%)	2,630,092 (6.76%)	3,220,527 (4.70%)	2,916,937 (4.26%)	4,322,642 (5.56%)	3,100,006 (6.04%)	4,257,415 (5.64%)	3,351,731 (5.59%)

CG1 and CG2 were males in the control group and CM1 and CM2 were females in the control group, while HG1 and HG2 were heat-stressed males and HM1 and HM2 were heat-stressed females.

## Data Availability

All the RNA-Sequencing data used in this study has been deposited in the Sequence Read Archive (SRA) public databases under BioProject (PRJNA531367).

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
