# Peer review of "Transcriptomic Analysis Provides Novel Insights into Heat Stress Responses in Sheep"

_animals, 2019, doi:10.3390/ani9060387_

Round 1
Reviewer 1 Report
This manuscript describes holistic approach to profile heat effects on liver expression in sheep. Sequencing data of liver transcripts found within liver of sheep from a heat-stressed group versus a control group are depicted.
The aim of this study is understandable but remains on an unsatisfying descriptive level. NGS results were diligently collected, analysed using suitable state-of-the-art software and discussed on a mainly theoretical level.
Important points of criticism are:
- poor description of animal selection breeding and keeping (were 36 animal selected once in August or twice in August as well as December?).
- did these sheep reach puberty?
- which part of the liver was collected (one may assume local expression differences)?
- at which time of the day were the samples taken - may circadian changes be responsible for the group differences in expression?
- were all RNA samples pool for each group or individual RNAs measured?
- for me astonishing is that no gender differences have been detected.
- when reporting expression changes the protein profiles must be presented for such transcript differences. Otherwise no biological effects can be predicted (it is well known that transcript changes will not automatically result in corresponding protein changes).
- when discussing important biological pathways (e.g. lipid metabolism) the matching blood parameters can be easily measured like NEFA, insulin, catecholamines, steroids, thyroid hormones etc.. Such surplus data are essential and must be added to this paper.
In general, all data sets generated from such animals/samples remain doubtful concerning additional influencing factors like length of the day, changing quality of the diets (no data presented), hormonal differences (steroids, thyroid gland), health stage of each animal etc..
Therefore, this paper remains on a medium preliminary scientific stage and must undergo a significant qualitative improvement. Functional and physiological effects have been only theoretically deduced based on weak datasets.
Author Response
Dear Reviewer:
RE: animals-498120
Title: Transcriptomic Analysis Provides Novel Insights into Heat Stress Responses in Sheep
Thank you for reviewing the above-referenced manuscript submitted earlier to your office. We would like to take this opportunity to express our appreciation to you. In accordance with the reviewers' comments and suggestions, the manuscript has been revised accordingly. We feel that this revised manuscript has been strengthened by the reviewers' comments and suggestions.
I hope the changes and explanations satisfy the requirements of the editorial board. I thank you again for reviewing the manuscript and look forward to hearing your favorable reply soon.
Yours sincerely,
Wei Caihong.
Reviewers' comments:
This manuscript describes holistic approach to profile heat effects on liver expression in sheep. Sequencing data of liver transcripts found within liver of sheep from a heat-stressed group versus a control group are depicted.
The aim of this study is understandable but remains on an unsatisfying descriptive level. NGS results were diligently collected, analysed using suitable state-of-the-art software and discussed on a mainly theoretical level.
1. Poor description of animal selection breeding and keeping (were 36 animal selected once in August or twice in August as well as December?).
Thank you. The experimental animals were selected once in August and December respectively, and 36 animals were selected for each time.
2. Did these sheep reach puberty?
Thank you. Hu sheep, as an excellent local variety in China, are characterized by rapid growth and reproductive capacity. Researches indicate that the weight of the 6-month-old lamb can reach to 87% of adult sheep (two years old). Besides, ewe generally matures at the age of 4-5 months and starts mating at 6 months old; however, ram generally starts mating at 8 months old. Therefore, the 5-month-old Huyang sheep selected in this study had reached puberty.
3. Which part of the liver was collected (one may assume local expression differences)?
Thanks. After the slaughter, the complete liver of sheep was separated promptly, then the left lobe of liver was selected for sample collection, and the collected part of each sheep was same approximately.
4. At which time of the day were the samples taken - may circadian changes be responsible for the group differences in expression?
Thanks. As an important metabolic organ of the body, the function exertion of liver might be influenced by diurnal change. Hence, all sheep in this study were slaughtered and sampled at empty stomach in the morning, so as to reduce sampling errors.
5. Were all RNA samples pool for each group or individual RNAs measured?
Thank you. In this study, 8 samples from heat stress group and control group were sequenced respectively.
6. For me astonishing is that no gender differences have been detected.
Thank you. The sheep from heat stress group and control group had different sexuality and gene. Thus the relevant analysis were carried out against group (CG1, CG2, CM1, CM2)vs(HG1, HG2, HM1, HM2). Furthermore, the results about group (CG1, CG2)vs(HG1, HG2) and group (CM1, CM2)vs(HM1, HM2) were demonstrated in the manuscript as attachment.
7. When reporting expression changes the protein profiles must be presented for such transcript differences. Otherwise no biological effects can be predicted (it is well known that transcript changes will not automatically result in corresponding protein changes).
Thank you. This was a good question. The mRNA abundance of a specific gene did not necessarily have a linear relationship with the protein expression quantity of its translation product. Certainly, the objective of this study was to research the genes and metabolic pathways of sheep that play a regulatory role in a heat stress environment. In subsequent studies, when the differentially expressed genes are required to be verified functionally, protein expression profiling will be taken into account to determine that some gene does differ between the two.
8. when discussing important biological pathways (e.g. lipid metabolism) the matching blood parameters can be easily measured like NEFA, insulin, catecholamines, steroids, thyroid hormones etc.. Such surplus data are essential and must be added to this paper.
Thank you. Related blood physiological and biochemical indexes were also measured, but these data were used in other articles (except for thyroid hormones). Therefore, thyroid hormone indexes were added into the manuscript.
Reviewer 2 Report
To unveil the underpinning molecular mechanisms related to sheep heat stress, the authors compared transcriptome data in livers from sheep harvested at August (Heat stress group) versus at December (Control group) and identified the 1137 DE genes. These results will shed light on further investigation of the heat stress related mechanism.
However, there are a few issues needed to be confirmed.
1 According to the general criteria of expressed transcript (FPKM >0.1), why the authors did mix none expressed genes with those low expression ones (FPKM values 
between 0–1)? 

2 The authors divided eight sheep into four groups including HG1, HG2, HM1, HM2, CG1, CG2, CM1 and CM2, while in Table 1 samples were named as RG1, RG2, RM1, RM2, besides CG1, CG2, CM1 and CM2?
3 Possibly sex is an important factor affecting heat response, therefore the authors sampled female and male sheep liver and sequenced them under both heat and control conditions. Nevertheless, why the authors did not describe the DE genes affected by sex?
4 Figure 4 the expression levels of gene quantified by RNA-seq or qRT-PCR were missing error bar.
5 Based only on the environment temperature of August, the authors suggested that sheep were in heat stress response is not persuaded enough. Could the authors provide more robust data to prove that these sheep were in heat stress indeed?
Author Response
Dear Reviewer:
RE: animals-498120
Title: Transcriptomic Analysis Provides Novel Insights into Heat Stress Responses in Sheep
Thank you for reviewing the above-referenced manuscript submitted earlier to your office. We would like to take this opportunity to express our appreciation to you. In accordance with the reviewers' comments and suggestions, the manuscript has been revised accordingly. We feel that this revised manuscript has been strengthened by the reviewers' comments and suggestions.
I hope the changes and explanations satisfy the requirements of the editorial board. I thank you again for reviewing the manuscript and look forward to hearing your favorable reply soon.
Yours sincerely,
Wei Caihong.
Reviewers' comments:
To unveil the underpinning molecular mechanisms related to sheep heat stress, the authors compared transcriptome data in livers from sheep harvested at August (Heat stress group) versus at December (Control group) and identified the 1137 DE genes. These results will shed light on further investigation of the heat stress related mechanism.
1.According to the general criteria of expressed transcript (FPKM >0.1), why the authors did mix none expressed genes with those low expression ones (FPKM values between 0–1)?
Thank you. Because of carelessness of author, (0.1–1) was written as (0–1), which has been modified correspondingly.
2. The authors divided eight sheep into four groups including HG1, HG2, HM1, HM2, CG1, CG2, CM1 and CM2, while in Table 1 samples were named as RG1, RG2, RM1, RM2, besides CG1, CG2, CM1 and CM2?
Thank you. Because of carelessness of author, (HG1, HG2, HM1, HM2) was written as (RG1, RG2, RM1, RM2), which has been modified correspondingly.
3. Possibly sex is an important factor affecting heat response, therefore the authors sampled female and male sheep liver and sequenced them under both heat and control conditions. Nevertheless, why the authors did not describe the DE genes affected by sex?
Thanks. Analysis results of differentially expressed genes affected by sexuality were added into the manuscript as attachment.
4. Figure 4 the expression levels of gene quantified by RNA-seq or qRT-PCR were missing error bar
Thanks. The error bars have been added in fig. 4 as requested.
5. Based only on the environment temperature of August, the authors suggested that sheep were in heat stress response is not persuaded enough. Could the authors provide more robust data to prove that these sheep were in heat stress indeed?
Thanks. T3 and T4 act in regulating the body’s metabolic heat production. Therefore, T3 and T4 concentrations in sheep serum from heat stress group and control group were detected, further illustrating that the sheep in this study were in a state of heat stress. Besides, the sheep houses THI of heat stress group and control group during the research were shown in attached fig. 1, which indicated that the heat-stressed sheep were subjected to extremely severe heat stress.
Round 2
Reviewer 1 Report
Most of the comments have been acknowledged. Problem remains the imbalanced animal experiment concerning puberty in ewes but pre-puberty in the rams.
Author Response
Thank you. Hu sheep, as an excellent local variety in China, are characterized by rapid growth and reproductive capacity. Researches indicate that the weight of the 6-month-old lamb can reach to 87% of adult sheep (two years old). Besides, ewe generally matures at the age of 4-5 months and starts mating at 6 months old; however, ram generally starts mating at 8 months old. Therefore, the 5-month-old Huyang sheep selected in this study had reached puberty [1].
In addition, the scientific questions initially designed in this study are: if under heat stress environment, which genes and metabolic pathways play a regulatory role in sheep? Are there any changes in fat metabolism? In this respect, we chose the same month-old sheep liver for related research. If ovaries or testicles are selected to study reproductive traits, the consistency of puberty between rams and ewes should be taken into consideration.
[1] Zhao, Y.Z. Chinese sheep production, 1st ed.; China agriculture press: Beijing, China, 2103; pp. 95–96.